# STOCHASTIC VISION TRANSFORMERS WITH WASSERSTEIN DISTANCE-AWARE ATTENTION

## ABSTRACT

Self-supervised learning is one of the most promising approaches to acquiring knowledge from limited labeled data. Despite the substantial advancements made in recent years, self-supervised models have posed a challenge to practitioners, as they do not readily provide insight into the model's confidence and uncertainty. Tackling this issue is no simple feat, primarily due to the complexity involved in implementing techniques that can make use of the latent representations learned during pre-training without relying on explicit labels. Motivated by this, we introduce a new stochastic vision transformer that integrates uncertainty and distance awareness into self-supervised learning (SSL) pipelines. Instead of the conventional deterministic vector embedding, our novel stochastic vision transformer encodes image patches into elliptical Gaussian distributional embeddings. Notably, the attention matrices of these stochastic representational embeddings are computed using Wasserstein distance-based attention, effectively capitalizing on the distributional nature of these embeddings. Additionally, we propose a regularization term based on Wasserstein distance for both pre-training and fine-tuning processes, thereby incorporating distance awareness into latent representations. We perform extensive experiments across different tasks such as in-distribution generalization, out-of-distribution detection, dataset corruption, semi-supervised settings, and transfer learning to other datasets and tasks. Our proposed method achieves superior accuracy and calibration, surpassing the self-supervised baseline in a wide range of experiments on a variety of datasets. Our code is in the supplementary material.

## 1 INTRODUCTION

Self-supervised representation learning has gained more importance in recent years owing to the expensive cost associated with obtaining real-life, well-labeled data. Through self-supervised learning, models learn to extract features solely from the latent representations of the data without the need for explicit labels. In recent years, self-supervised learning techniques have demonstrated state-of-the-art performance in a wide range of tasks including natural language processing (NLP; (Devlin et al., 2018; Brown et al., 2020)), computer vision(Chen et al., 2020; Bardes et al., 2021a; Grill et al., 2020), as well as multimodal learning (Radford et al., 2021; Li et al., 2022; Shi et al., 2022). However, most of the existing approaches do not provide adequate information on the models' confidence and reliability in both pretext learning and downstream tasks, therefore potentially yielding unreliable, overconfident performance. Thorough estimation and investigation of a model's output provide insight into the model's confidence and potential fail cases when the models are uncertain of their predictions. In light of the potential crucial real-life deployment of self-supervised learning, the development of methods that ensure reliable and safe self-supervised learning frameworks is a crucial yet non-trivial task. We adhere to Plex's (Tran et al., 2022) definition of reliability, which evaluates a model's capability to consistently perform well across a variety of tasks. Specifically, Tran et al. (2022) introduces three key criteria for reliable machine learning systems: the model should demonstrate robust generalization to *new tasks*, effectively adapt to *new datasets*, and faithfully represent the corresponding *uncertainty*.

In literature, Bayesian neural networks (BNNs; Neal (2012)) and neural ensemble networks (Hansen & Salamon, 1990)(Lakshminarayanan et al., 2017) are popular and widely used methods for capturing parameter distributions in machine learning. While the Bayesian paradigm offers a principled

approach to uncertainty quantification, it is not ideal for self-supervised methods. Bayesian deep learning, which typically involves sampling from the posterior distribution, faces scalability challenges with the prevalent large architectures in this field and relies heavily on having true labels $y$ for maximum-a-posterior (MAP) estimation. One possible solution for self-supervised learning is to inject stochasticity directly into the encoder networks. In this manner, the models return a diverse set of solutions through appropriate randomization of the weights, analogous to the randomization of weights achieved through Deep Ensembles and Monte-Carlo (MC)-Dropout (Gal & Ghahramani, 2016), without training multiple sets of large models.

In this paper, we formalize a comprehensive method for robustness and distance-aware self-supervised learning through stochastic vision transformer encoders. Our novel method ensures more robust predictions with a negligible decrease in predictive performance.

We summarize our contributions as follows:

1. We propose an alternative stochastic transformer architecture with distributional embedding for masking-based vision SSL. We introduce stochasticity into the attention mechanism through a Wasserstein distance-based attention mechanism, which determines the stochastic attention matrix between the embedded distributions.

2. We introduce novel Wasserstein distance-based regularization terms in the loss objectives for both unsupervised pre-training and supervised downstream tasks. The regularization terms leverage uncertainty and distance awareness into the training, encouraging similar items to be embedded closer together in the distributional space.

3. We perform comprehensive experiments to show the advantage of our method compared to deterministic approaches. We test our method's predictive and uncertainty measurements in 1) In-distribution tasks; 2) Out-of-distribution tasks; 3) Distribution shift via image corruption tasks; and 4) Semi-supervised learning tasks; Our approach achieves superior downstream predictive performance to uncertainty trade-off compared to the other methods, which highlights the promising benefit of distance-aware training to self-supervised learning frameworks.

## 2 RELATED WORKS

**Self-supervised Learning** Due to the potential scarcity of labels in real-world applications, self-supervised learning offers a feasible advantage over fully supervised learning as features are pre-learned directly from the latent representation of the data during pre-text tasks without explicit annotation. Current established self-supervised learning methods involve solving pre-text tasks, ranging from masking-based data reconstruction (Baevski et al., 2022), contrastive learning (Caron et al., 2021a; Chen et al., 2020; Zbontar et al., 2021), contrastive divergence learning (Rezaei et al., 2021), to knowledge distillation (Caron et al., 2021b; Vahidi et al., 2023a). The refinement of the learned representations occurs in the subsequent downstream tasks, which, in most cases, involve classic fully supervised learning tasks, such as visual object classification and sentiment analysis. In some use cases such as anomaly detection (Bozorgtabar et al., 2021; Tran et al., 2022) or out-of-distribution detection (Mohseni et al., 2020; Khalid et al., 2022; Vahidi et al., 2023b;a), the features learned from pre-text tasks are directly used for inference. Previous works have sought to improve the robustness of the pre-text of learning by tackling representation collapse (Bardes et al., 2021b; Rezaei et al., 2023) or explicitly improving the models' OOD performance (Winkens et al., 2020; Sehwag et al., 2021; Rezaei et al., 2022; Tran et al., 2022). Nevertheless, the concepts of robustness and distance awareness of self-supervised learning frameworks have not been investigated to a significant degree.

**Transformer Uncertainty Estimation** One possible way to inject uncertainty awareness within self-supervised networks is to inject stochasticity into the encoders. In the case of transformer encoders, recent works have injected stochasticity into the self-attention mechanism of the transformer using Gumbel softmax (Pei et al., 2022), double stochastic attention matrix with Sinkhorn algorithm (Sander et al., 2022), or Gaussian mixture model (Nguyen et al., 2022). While (Pei et al., 2022) investigates the uncertainty estimation of its hierarchical stochastic transformer method in fully supervised NLP tasks, no rigorous study on uncertainty estimation of the aforementioned stochastic transformers in self-supervised learning networks has been performed. The stochastic transformer

for recommendation system (Fan et al., 2022) has also implemented distributional embeddings, Wasserstein distance-based attention mechanism, and loss term into the classical transformer, which was nevertheless mainly developed to improve the predictive performance of the fully supervised recommendation system tasks.

## 3   BACKGROUND AND PROBLEM FORMULATION

**Attention Mechanism** Transformers are developed as a competitive alternative to Recurrent Neural Networks (RNN) for sequential data, enabling long-term dependencies of the data sequence in place of short-term dependencies in RNNs. The key component of transformers is the attention mechanism, which determines contextual correlations between the embedded vector components within each batch. The input data is first tokenized into tokens of a given sequence length $l$ and vector dimensions $d$. Within each head of the multi-head attention of head size $h$, input data token $\boldsymbol{x} \in \mathbb{R}^{l \times d}$ is linearly projected into three vectors, namely query $Q \in \mathbb{R}^{l \times h \times \frac{d}{h}}$, key $K \in \mathbb{R}^{l \times h \times \frac{d}{h}}$, and value $K \in \mathbb{R}^{l \times h \times \frac{d}{h}}$. Attention is calculated by applying a scaled dot product to each element of the query vector with each element of the key vector. The obtained scaled dot product is then subsequently normalized with a Softmax function and multiplied with the value vectors. The self-attention mechanism can be summarized as follows:

$$\text{Attention}(Q, K, V) = \text{softmax}\left(\frac{QK^T}{\sqrt{d}}\right) V \tag{1}$$

$$Q = W_Q \boldsymbol{x}, K = W_K \boldsymbol{x}, V = W_V \boldsymbol{x}; \quad W_Q, W_K, W_V \in \mathbb{R}^{d \times d} \tag{2}$$

Every component of the classical attention mechanism is deterministic, returning a single dot vector output, thus rendering uncertainty estimation impossible. Therefore, we seek to introduce stochasticity into the attention mechanism by applying Wasserstein distance in place of the dot product.

**Probabilistic Embedding** The conventional transformer embeds input data tokens into vector points. Positional encoding is applied to maintain the positional information of the embedded vectors. Again, the embedding components found in classical transformers are deterministic. An alternative approach is to map the data tokens into probability distributions instead of point vectors. The embedding of text data into Gaussian distributions has been explored by Vilnis & McCallum (2015) and Qian et al. (2021), with the advocated benefit of uncertainty representation in the embedding space and diversity of representation in comparison to the dot product. Gaussian distribution embedding is also applied within the stochastic transformer for recommendation system sequences (Fan et al., 2022). Inspired by these previous works, we map image tokens into elliptical Gaussian distributions by encoding both mean and covariance vectors into the network. This paper introduces additional positional embeddings for both the mean and covariance embeddings. The Elliptical Gaussian distribution is chosen as it is relatively easy to parameterize and model in higher dimensions. Additionally, closed-form solutions exist for statistical inference and measure operations with Gaussian distributions, allowing computationally and methodically simpler stochastic implementations.

**Wasserstein Distance in Machine Learning** Due to its empirical success and relative ease of implementation, the Wasserstein distance is one of the most commonly used distance metrics employed in machine learning applications which involve the learning of probability distributions. Given the formulation of the $p$-Wasserstein distance with $p \geq 1$ in Eq. 3, calculating the $p$-Wasserstein distance may be intractable for large $p$ and arbitrary distribution functions. However, a tractable closed-form solution exists for the case of 2-Wasserstein distance with Gaussian distribution functions, formulated as follows (Dowson & Landau, 1982)(Olkin & Pukelsheim, 1982)(Knott & Smith, 1984)(Givens & Shortt, 1984),

$$\mathcal{W}_p(\mu, \nu) = \left( \inf_{\gamma \in \Gamma(\mu, \nu)} \int_{\mathcal{X} \times \mathcal{X}} c(x, y)^p d\gamma(x, y) \right)^{1/p} \tag{3}$$

$$W_2^2(z_1, z_2) = \left\| \mu_1 - \mu_2 \right\|^2 + Tr(\Sigma_1 + \Sigma_2 - 2(\Sigma_1^{1/2} \Sigma_1 \Sigma_2^{1/2})^{1/2}), \tag{4}$$

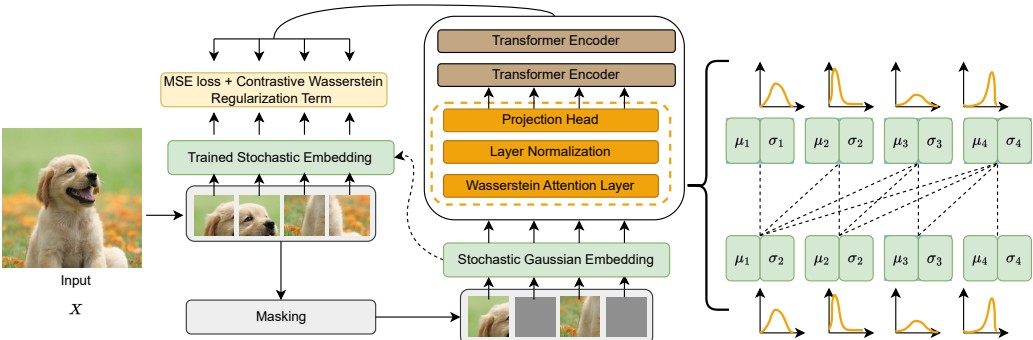

Figure 1: Illustration of our proposed pre-training pipeline. Given a mini-batch of $\boldsymbol{X}$ input sample, the augmented view of $\boldsymbol{x}'$ and the original view $\boldsymbol{x}$ are encoded into a set of features by the stochastic encoder network to produce robust representations via stochastic elliptical Gaussian embeddings.

for Gaussian probability measures $z_1 \sim \mathcal{N}(\mu_1, \Sigma_1)$ and $z_2 \sim \mathcal{N}(\mu_2, \Sigma_2)$ on $\mathbb{R}^d$. $Tr$ denotes the trace operator of the covariance matrices. While the Wasserstein distance has been essential in various machine learning applications Arjovsky et al. (2017), the use of Wasserstein distances in improving robustness and correlating distributional embeddings has not been explored in detail. Distributionally robust optimization (Gao et al., 2020; Kuhn et al., 2019) has empirically demonstrated the potential to capture uncertainty and improves the robustness of neural networks. We build upon these previous works and introduce the tractable formulation of the 2-Wasserstein distance into transformers' attention layer and an additional regularization term. Our main objective is to leverage the distances between the stochastic Gaussian embeddings into the main learning process of neural networks, facilitating more robust learning.

## 4 METHOD

Given a mini-batch of training data, randomly sampled and defined as $\boldsymbol{X} = [\boldsymbol{x}_1, \ldots, \boldsymbol{x}_n]^N \in \mathbb{R}^{N \times D}$ and transformation function $\tau$ that operates on this data. The transformation function plays a crucial role in improving the training process by generating an augmented view, $\tilde{\boldsymbol{x}} \triangleq \tau(\boldsymbol{x})$ for each sample in $\boldsymbol{X}$. The augmentation process involves sampling $\tau$ from a distribution of suitable data transformations. Examples of such transformations include partially masking image patches (He et al., 2022) or applying various image augmentation techniques (Chen et al., 2020). Later, we create an image token using a pre-trained convolutional neural network (CNN) to extract features from each image and then aggregate these features into a token representation $\boldsymbol{t}_0^0, ..., \boldsymbol{t}_0^L$ for each image in the batch similar to (Dosovitskiy et al., 2021). As depicted in Figure 1, the image token is generated both for the original input data and for the augmented samples. Differing deterministic vector representation, we create two vectors of mean $\mu$ and variance $\sigma$ based on a sequence of tokens of augmented samples $\tilde{\boldsymbol{x}}$ which are fed to our proposed stochastic Gaussian embedding layer.

### 4.1 STOCHASTIC GAUSSIAN EMBEDDING

We represent tokens in the stochastic Gaussian embedding space with mean $\mu$ and variance $\sigma$ vectors, thereby instilling distributional information throughout the entire architecture. In contrast to deterministic embedding representations, we append two separate positional encoding vectors for $\mu$ and $\sigma$, forming the stochastic Gaussian embeddings $\boldsymbol{z}_\mu^0, ..., \boldsymbol{z}_\mu^L$ and $\boldsymbol{z}_\sigma^0, ..., \boldsymbol{z}_\sigma^L$ which are then passed to the stochastic transformer encoder blocks. Each encoder block comprises normalization layers, stochastic Wasserstein attention, and a projection head.

**Wasserstein Attention** We propose an attention mechanism based on the Wasserstein distance, enabling effective focus on stochastic Gaussian embeddings. The stochastic embeddings undergo linear transformations, forming stochastic Gaussian Q(Query), K(Key), and V(Value) representations $\boldsymbol{z}_{qkv} \sim N(\mu_{qkv}, \sigma_{qkv})$, formulated as follows,

$$\mu_{qkv} = z_\mu W_{qkv}^\mu$$
$$\sigma_{qkv} = \mathbf{ELU}(diag(\boldsymbol{z}_\sigma W_{qkv}^\sigma)) + 1. \tag{5}$$

The ELU activation function preserves the positive definite property of the covariance matrix. By evaluating the negative 2-Wasserstein distance instead of using dot-product attention between the stochastic Gaussian $Q$ and $K$ embeddings, we derive the attention scores. This is formulated as follows:

$$A_{Q,K} = -(W_2^2(Q,K)) = -(\left\|\mu_Q - \mu_K\right\|^2 + Tr(\Sigma_Q + \Sigma_K - 2(\Sigma_Q^{1/2}\Sigma_Q\Sigma_K^{1/2})^{1/2})), \tag{6}$$

$$A_{\boldsymbol{z}} = \text{softmax}\left(\frac{A_{Q,K}}{\sqrt{d}}\right). \tag{7}$$

We obtain the stochastic Wasserstein embedding by multiplying the attention scores with the corresponding stochastic value embeddings as in Equation 8 and Equation 9. Repeating this calculation across the block depth ensures that the weights comprehensively learn the spatial correlation of the embedded stochastic distributions through the evaluated distances between the Gaussian distributions.

$$A_\mu = A_{\boldsymbol{z}} V_\mu, \tag{8}$$
$$A_\sigma = A_{\boldsymbol{z}}^2 V_\sigma. \tag{9}$$

## 4.2 Contrastive Wasserstein Regularization

We formulate a contrastive Wasserstein distance-based regularization term that enables the learning of distances between the embedded stochastic Gaussian embedding. The main deterministic training objective remains in place, thereby ensuring a comprehensive learning of the representations while also encoding robustness into the objective. Given the contrastive nature of the regularization term, we distinguish the case between unsupervised pre-training and supervised fine-tuning, whereby positive and negative example pairs are to be considered. Consider one mini-batch of the output embeddings returned from the stochastic transformer encoder $\boldsymbol{z}_{out} \sim N(\mu_{out}, \sigma_{out})$, the trained stochastic embedding layer $f_z$, and the positive example $y^+$. The cumulative loss term during pre-training of a self-supervised learning framework is given as follows:

$$L_p = \mathcal{L}_p - \lambda \log(\sigma(-W_2^2(\boldsymbol{z}_{out}, f_z(y^+)))), \tag{10}$$

where $\mathcal{L}_p$ is the deterministic unsupervised pre-training loss term and $W_2^2$ is the 2-Wasserstein operation. We consider the unmasked image patches as the positive unsupervised pre-training example. The 2-Wasserstein operation encourages the network to minimize the distance between the stochastic representations of the masked and unmasked image patches. We control the magnitude of the regularization term with the parameter $\lambda$. Similarly, the fine-tuning objective with the corresponding regularization parameters $\lambda_1$ and $\lambda_2$, is given as follows:

$$L_f = \mathcal{L}_{CE} - \lambda_1 l_1(\boldsymbol{z}_{out}, y^+, y^-) + \lambda_2 l_2(\boldsymbol{z}_{out}, y^+, y^-), \tag{11}$$

whereby the distributional regularization terms $l_1$ and $l_2$ are given as follows:

$$l_1(\boldsymbol{z}_{out}, y^+, y^-) = \log(\sigma(W_2^2(\boldsymbol{z}_{out}, f_z(y^+)) - W_2^2(\boldsymbol{z}_{out}, f_z(y^-)))), \tag{12}$$

$$l_2(\boldsymbol{z}_{out}, y^+, y^-) = [W_2^2(\boldsymbol{z}_{out}, f_z(y^+)) - W_2^2(f_z(y^+), f_z(y^-))]_+. \tag{13}$$

$\mathcal{L}_{CE}$ is the supervised cross-entropy loss term, $[x]_+ = \max(x, 0)$ is the hinge loss operator, $y^+$ and $y^-$ are the positive and negative examples. The extended regularization terms correspond to the comprehensive contrastive learning from both positive and negative examples. The term $l_1$ regularizes the distance between the stochastic embeddings of the input data and the examples, while $l_2$ enforces a larger distance between the positive and negative examples. We consider the unaugmented image patches to be positive examples. The access to ground truth labels facilitates the explicit random sampling of images belonging to other classes which serve as negative examples.

## 5 EXPERIMENTAL SETTINGS

We conducted several experiments to assess the robustness of our stochastic transformer method during both pretext and downstream tasks.

**Network Architecture** Our implemented method builds upon the data2vec self-supervised learning framework (Baevski et al., 2022) with a ViT backbone (Dosovitskiy et al., 2020). Data2vec is a masking-based framework that enables the backbone networks to acquire knowledge by reconstructing latent representations of masked data using the latent representations of the original, unmasked data. In the pretext task, our primary learning objective involves minimizing the smoothed L1 loss of the reconstruction of the masked data while the cross-entropy loss is used during the downstream tasks. For our experiments, we chose to train with the ViT-B backbone. In addition, we introduced regularization terms to enhance the robustness of the learning process, as explained in the previous section.

**Optimization** The hyperparameters and model parameters optimized are in line with the values used for both pretext and downstream training of the data2vec framework for the imaging modality unless otherwise stated. For the pretraining, we optimize the network with a total batch size of 1024, a learning rate of $2 \times 10^{-3}$, and the stochastic regularization term $\lambda$ of $1 \times 10^{-5}$ for a total of 300 epochs. For the downstream task, we optimize the network with a total batch size of 512, a learning rate of $5 \times 10^{-3}$, the stochastic regularization terms $\lambda_1$ of $1 \times 10^{-4}$ and $\lambda_2$ of $1 \times 10^{-4}$ for 50 epochs.

**Dataset** We conducted our experiments using the following datasets: **CIFAR-100**and **CIFAR-10** (Krizhevsky, 2009) : These datasets consist of tiny images with dimensions of 32 x 32 featuring 100 and 10 distinct classes, respectively. The images are split into 50,000 training images and 10,000 validation images for both datasets. **SVHN** (Street View House Numbers) (Netzer et al., 2011): This dataset is composed of 600,000 RGB 32 x 32 images of 10 classes of house numbers ( 0 to 9 ) taken from Google Street View. **CIFAR-100-C** and **CIFAR-100-P** (Hendrycks & Dietterich, 2019) are commonly used benchmarks to evaluate a model's robustness against corruptions and perturbations, respectively. CIFAR-100-C consists of CIFAR-100 images that have been subjected to 18 distinct types of image corruption each at five different severity levels. On the other hand, CIFAR-100-P includes sequences of image frames from CIFAR-100 with specific perturbation applied progressively over the time frames.

**Tasks** We assess the effectiveness of our approach across various tasks considering evaluation protocols by self-supervised learning (Chen et al., 2020) and robustness evaluations suggested by Plex (Tran et al., 2022). In particular, we evaluate the model's performance on In-Domain generalization (IND), Out-Of-Distribution (OOD) detection, semi-supervised learning, and corrupted and perturbed dataset robustness tasks.

**Evaluation Metrics** We report the performance metrics using the following notation: upward arrows signify that higher values are considered more optimal while downward arrows indicate the opposite. **Top-1 Accuracy** $\uparrow$: the number of correct top-1 class predictions for a given test sample batch. Top-1 Accuracy measures the predictive performance of the model. **AUROC** $\uparrow$: Area Under Receiver Operating Characteristic curve. This metric measures the model's ability to differentiate positive and negative classes. For OOD detections, a sample has a negative class if it is OOD and a positive class otherwise. **NLL** $\downarrow$: the negative log-likelihood of the predicted distribution given the true target values. NLL measures the difference between the predicted confidence and true confidence, with lower values indicating that the model is better at predicting the true labels. **ECE** $\downarrow$: measures the discrepancy between accuracy and confidence scores. A lower ECE value signifies better-calibrated models, returning higher accuracy when they are more confident and vice-versa. **mCE** $\downarrow$: used in corrupted dataset evaluations. Previous works defined mCE as the average of corruption errors over several corruption types and severity (Wang et al., 2022)(Hendrycks et al., 2020). **MFP** $\downarrow$: measures the probability that two adjacent perturbation sequence frames of the same image result in a flip of two distinct output classes in perturbed dataset evaluation. (Hendrycks & Dietterich, 2019). **Top-5 distance** $\downarrow$: measures the stability of top-5 predictions across perturbation sequence (Hendrycks & Dietterich, 2019).

**Compared Methods** We benchmark our method against the following competing approaches. **Baseline** Baseline self-supervised data2vec framework (Baevski et al., 2022) for image modal-

ity . **Deep Ensembles** We consider deep ensembles with 10 random seeds of the baseline networks. **MC-Dropout** Baseline encoders with dropout regularization applied during pre-training, fine-tuning, and inference with 10 forward passes. **Sinkformer** Baseline networks with stochastic Sinkhorn algorithm-based transformer architecture applied during both pre-training and fine-tuning. **SNGP** Baseline networks with spectral normalization and Gaussian process layers instead of linear layer applied during fine-tuning. The results shown were averaged over 5 runs.

## 6   RESULTS AND DISCUSSION

 **In-Distribution Generalization** For the in-distribution generalization evaluation, we measure the top-1 accuracy and the calibration error of the methods detailed in the previous section. We pre-train and fine-tune the networks with the training set. The top-1 accuracy and calibration errors are obtained from inference with the test set. An ideal solution offers high predictive performance while maintaining low calibration errors. Table 1 summarizes the evaluation results for both CIFAR-100 and CIFAR-10 datasets. In terms of ECE and NLL, our method outperforms the Deep Ensembles and Baseline methods. Moreover, our method achieves a higher top-1 accuracy value in comparison to the other uncertainty quantification methods and stochastic transformer methods. The results suggest that our method leads to more robust and generalizable self-supervised learning predictions without sacrificing the prediction accuracy.

Table 1: Results of In-Distribution accuracy and calibration error for our stochastic embedding networks trained with CIFAR-100/10. The best score for each metric is shown in **bold**, and the second-best is underlined

| Methods | CIFAR-100 | | | CIFAR-10 | | |
|---|---|---|---|---|---|---|
| | **Acc** (↑) | **NLL** (↓) | **ECE** (↓) | **Acc** (↑) | **NLL** (↓) | **ECE** (↓) |
| Baseline | **69.720** | **1.198** | 0.456 | **76.933** | 0.816 | 0.415 |
| Ensembles k = 10 | 69.040 | 1.229 | 0.454 | 76.470 | 0.789 | 0.402 |
| Sinkformer | 59.360 | 1.591 | 0.474 | 56.970 | 1.327 | 0.437 |
| MC-Dropout 0.3 | 54.180 | 1.834 | 0.450 | 73.086 | 0.905 | 0.429 |
| SNGP | 64.896 | 1.411 | 0.487 | 74.580 | 0.894 | 0.437 |
| Our method | 69.420 | 1.223 | **0.445** | 76.630 | **0.784** | **0.397** |

**Out-of-Distribution Detection** OOD predictions gauge the models' capability to identify unseen test samples stemming from different distributions. We pre-train and fine-tune the models with the ID training datasets and perform zero-shot OOD inference with OOD training datasets. Table 2 summarizes the OOD detection experiment results for differing datasets and methods. From the AUROC values obtained, our method outperforms the other competing methods, notably the distance-aware SNGP, highlighting the importance of the distance-aware regularization and distributional embeddings to the out-of-distribution robustness of the networks.

Table 2: Results of Out-of-distribution AUROC for our stochastic embedding networks trained with CIFAR-100/10. The best score for each metric is shown in **bold**, and the second-best is underlined.

| Methods | CIFAR-100 ID | | CIFAR-10 ID |
|---|---|---|---|
| | **CIFAR-10** (↑) | **SVHN** (↑) | **SVHN** (↑) |
| Baseline | 0.519 | 0.483 | 0.483 |
| Ensembles k = 10 | 0.518 | 0.492 | 0.487 |
| Sinkformer | 0.559 | 0.489 | 0.462 |
| MC-Dropout 0.3 | 0.485 | **0.512** | 0.480 |
| SNGP | 0.584 | 0.505 | 0.486 |
| Wasserstein | **0.629** | 0.498 | **0.490** |

**Corrupted and Perturbed Dataset Evaluation** Given the stochastic nature of real-life observations, it is crucial for models to infer robust predictions from data affected by distribution shifts, for example through noisy images. The corrupted and perturbed dataset experiments emulate the possible real-life distribution shifts induced by noise and distortions. We pre-train and fine-tune the models with the in-distribution datasets. We then perform zero-shot inference with the corrupted

Table 3: Results of the CIFAR-100-C/CIFAR-100-P test dataset inference. The best score for each metric is shown in **bold**, and the second-best is underlined.

| Methods | CIFAR-100-C mCE($\downarrow$) | CIFAR-100-P MFP ($\downarrow$) | Top-5($\downarrow$) |
|---|---|---|---|
| Baseline | 0.506 | 14.294 | 2.647 |
| Ensembles k = 10 | 0.505 | 12.815 | **2.394** |
| Sinkformer | 0.581 | 16.654 | 3.072 |
| MC-Dropout 0.3 | 0.602 | 14.235 | 2.782 |
| SNGP | 0.527 | 15.211 | 2.680 |
| Our method | **0.487** | **12.808** | 2.410 |

Table 4: Semi-supervised classification accuracy for both baseline data2vec and stochastic Wasserstein transformer data2vec networks. The best score for each metric is shown in **bold**.

| Methods | Label fraction 1% | | | 10% | | |
|---|---|---|---|---|---|---|
| | Top-1 | ECE | NLL | Top-1 | ECE | NLL |
| Baseline | **22.260** | 0.287 | 3.091 | 56.980 | 0.466 | 1.676 |
| Wasserstein | 21.710 | **0.283** | **3.086** | **57.860** | **0.462** | **1.578** |

CIFAR-100/10-C dataset and the corrupted CIFAR-100/10-P dataset. Our findings are illustrated in Table 3. The lower CE Error and Mean Flip Probability values substantiate the improved robustness and generalization ability of the data2vec network applied with our method. While the ensemble method performs better in terms of the Top-5 distance metric, our method's performance is almost on par with the deep ensembles. The distributional embeddings and distance-aware regularization facilitate more robust predictions under distribution shifts.

**Semi-Supervised Learning** In semi-supervised learning, models are trained in low data regimes, emulating the possible real-life case of labeled data scarcity. We freeze the encoders of the networks that are pre-trained with the training set and subsequently fine-tune the linear classifier with 1% and 10% of the training set. The resulting top-1 accuracy, ECE, and NLL values from Table 4 indicate that our method facilitates more optimal and robust training in low data regimes.

## 7 ABLATION STUDIES

To develop a better understanding of the behavior and observed performance of our proposed method, we conducted a series of ablation studies to explore various aspects of our approach. These studies included (i) the analysis of computational cost compared to baseline and model ensemble, (ii) the impact of data augmentation, (iii) the impact of hyperparameters such as mini-batch size and number of epochs, as well as (iv) the impact of hyperparameters of our proposed regularization term.

**Analysis of computational cost** We evaluated the efficiency of our proposed method and compared it with baseline and deep ensemble in terms of the number of parameters, memory usage, and training time for 300 epochs. The results obtained in Table 5 show the SSL-Ensemble method leads to a notable increase in both memory and computational demands compared to the baseline while our method exhibits better efficiency in terms of a number of parameters, memory usage as well as training time.

Table 5: Computational Cost in 8 DGX-A40 60G GPUs on CIFAR100.

| Methods | Members | Parameters | Memory/GPU | Time / 300-ep |
|---|---|---|---|---|
| Baseline(SSL) | 1 | 86.3 M | 31.2 G | 4.0 (h) |
| SSL-Ensemble | 10 | 10 × 86.3 M | 10 × 31.2 G | 10 × 4.0 (h) |
| Our method | 1 | 115.8 M | 42.3 G | 9.3 (h) |

**Impact of augmentation magnitudes and severity**   We analyze the influence of augmentation magnitudes and severity by adjusting the *RandAugment* augmentation policy applied to our networks. Our findings are summarized in Table 7. Higher magnitude and more severe augmentation policies enforce a stronger contrastive regularization term, in alignment with the expected property of a contrastive loss term. This finding further highlights the importance of balancing the main deterministic loss objective and the contrastive regularization term.

**Impact of hyperparameters**   The influence of pre-training batch size and training epochs on the downstream performance is summarized in Figure 2. We observe that a smaller pre-training batch size leads to a generally better performance. Meanwhile, a batch size of 512 impedes the learning of representations, while a batch size of 1024 leads to a more satisfactory performance. We believe that the learning of representations is determined by the interplay between the deterministic loss term and the stochastic regularization loss term. For masked token reconstruction-based SSL networks, large batch sizes generally lead to improved unsupervised pre-training. However, with bigger batch sizes, the contrastive regularization term dominates the overall training loss, in alignment with the expected property of a contrastive loss term.

**Impact of regularization parameters**   Table 6 shows that appropriate tuning of the contrastive regularization coefficients $\lambda_1$ and $\lambda_2$ is crucial to the stability and performance of our stochastic Wasserstein network. Excessively high or low values impede the model's capacity to learn. A large coefficient value results in the contrastive regularisation term taking precedence over the training process, while a small coefficient value causes unstable training.

Table 6: Performance of Wasserstein attention transformers with differing stochastic regularization parameters.

| $\lambda_1$ | $\lambda_2$ | **Accuracy**($\uparrow$) |
|---|---|---|
| $1e^{-4}$ | $1e^{-4}$ | **69.420** |
| $1e^{-3}$ | $1e^{-3}$ | 61.310 |
| $1e^{-4}$ | $1e^{-3}$ | 60.460 |
| $1e^{-3}$ | $1e^{-4}$ | 59.390 |
| $1e^{-5}$ | $1e^{-5}$ | 51.160 |

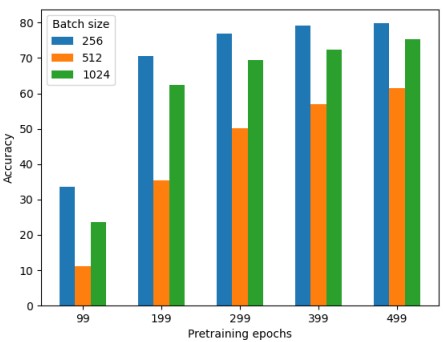

Figure 2: Linear ablation evaluation of the robust stochastic transformer pre-trained with different batch sizes and epochs. Each bar is a single training run.

Table 7: Performance of Wasserstein attention transformers with differing augmentations magnitude and amount. The augmentations are constructed with the RandAugment augmentation policy.

| Magnitude | Amt. Aug | **Accuracy** ($\uparrow$) |
|---|---|---|
| 9 | 2 | 69.420 |
| 9 | 4 | 58.780 |
| 9 | 1 | **71.370** |
| 10 | 2 | 66.560 |
| 6 | 2 | 70.920 |

## 8   Conclusion

 In this paper, we presented stochastic Gaussian embeddings for learning robust representation of self-supervised framework. Specifically, we introduced a stochastic Wasserstein distance-based attention mechanism to attend to the embedded tokens, passing down the stochastic information in the process. The Wasserstein-distance-based regularization term is proposed to leverage the distance between embedded output distributions into the learning process. Our distance-aware stochastic implementation encourages more robust self-supervised learning performance, as evident from the promising in-distribution generalization, OOD detection, corrupted dataset evaluation, and semi-supervised learning evaluation in comparison to other well-established methods. In future work, we will explore our stochastic embedding implementations with other data modalities. Additionally, we will perform intensive investigations into the information geometry aspect of the embedded data, alternative distances, and numerical approximations which are optimized for the stochastic embedding of higher dimensional data.

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
