# OpenReview forum: "Stochastic Vision Transformers with Wasserstein Distance-Aware Attention"
_ICLR.cc/2024/Conference — Submitted to ICLR 2024_

### Official Review · Reviewer_ysA7 · 2023-10-30

**Soundness:** 3 good
**Presentation:** 3 good
**Contribution:** 2 fair
**Rating:** 5
**Confidence:** 2

**Summary:**

The paper adopts a stochastic transformer architecture that uses distributional embedding and Wasserstein distance-based attention mechanism for self-supervised learning pipelines. The contrastive regularization terms are added to the training objective.

**Strengths:**

- The paper is well-written and concise.
- The paper investigates an important research question, which can potentially interest many researchers.

**Weaknesses:**

- More explanations needed for baselines. e.g. For the baseline MC-Dropout, it only mentions the dropout regularization applied during pre-training, and the ratio is set to 0.3; but for MC dropout, the dropout is applied at both training and test time, and there is no detail for the testing.
- The performance is not convincing enough. The improvement is not significant, and according to the ablation study, it also seems very sensitive to the hyperparameters.
- It's not proper for the paper to say 'We propose a stochastic transformer architecture with distributional embedding......', since the distributional embeddings and Wasserstein distance-based attention mechanism are the same as Fan et al., 2022.

**Questions:**

Why set dropout ratio as 0.3 for the MC-Dropout baseline? Isn't it slightly higher than the common options?

---

> ### Author Response · Authors · 2023-11-20
> **Response to Reviewer ysA7**
>
> We thank the reviewers for the constructive comments!
> ## Explanations of baselines
> For MC-Dropout, we applied dropouts for pretraining, finetuning, and testing. We have omitted these details because of limited space in the paper. We have now detailed this in the new version of our paper. For the testing of the baseline MC-Dropout, we perform multiple ( k = 10 ) forward inference passes with the same dropout rate applied and subsequently average the outputs from all the forward runs.
>
> ## Significance of the results
> > The performance is not convincing enough. The improvement is not significant, and according to the ablation study, it also seems very sensitive to the hyperparameters.
> Our paper highlights the potential of the stochastic attention method as compared to other well-established uncertainty estimation methods, namely Deep Ensembles and MC-Dropout. As far as we understand, there are no other uncertainty estimation methods applicable to transformer-based SSL and our method outperforms the current state-of-the-art Deep Ensembles method. The main intent of our paper is to highlight the potential of the stochastic distributional attention mechanism for uncertainty estimation in applications where labels are not readily available. Regarding the sensitivity to hyperparameters, this is a common trait in advanced models in our domain, including the baselines. Despite this, our method outperforms well-established baselines across various settings, indicating robustness.
>
> ##
> > It's not proper for the paper to say 'We propose a stochastic transformer architecture with distributional embedding......', since the distributional embeddings and Wasserstein distance-based attention mechanism are the same as Fan et al., 2022.
>
> Thank you for highlighting this statement. This statement should be rephrased as
> > “We propose an alternative stochastic transformer architecture with distributional embedding for masking based vision SSL”
>
> as we did not intend to claim the architecture and embedding but rather its application for uncertainty quantification in SSL domain. We would again like to point out that the modifications we performed to accommodate the higher embedding dimension of image data in vision transformers and the stochastic regularization property of the distributional embedding in contrast to the implementation done by Fan et al (2022).
>
> ## Question regarding dropout rate
> >Why set dropout ratio as 0.3 for the MC-Dropout baseline? Isn't it slightly higher than the common options?
>
> We chose a dropout rate of 0.3 taking into account the dropout rates used in our reference papers. Pei et al. (2021) ([1]) used dropout rates of 0.5/0.1 for their hierarchical stochastic transformer uncertainty estimation. Miani et al. (2022) ([2]) used a dropout rate of 0.2 for their unsupervised representation learning uncertainty estimation method.
>
> [1] Pei, J., Wang, C., & Szarvas, G. (2021). Transformer uncertainty estimation with hierarchical stochastic attention. (AAAI 2022)
>
>
> [2] Miani, M., Warburg, F., Moreno-Muñoz, P., Detlefsen, N. S., & Hauberg, S. (2022). Laplacian autoencoders for learning stochastic representations. (NeurIPS 2022)

---

### Official Review · Reviewer_67qY · 2023-10-31

**Soundness:** 3 good
**Presentation:** 3 good
**Contribution:** 2 fair
**Rating:** 5
**Confidence:** 5

**Summary:**

The paper introduces a stochastic vision transformer that integrates uncertainty and distance awareness into self-supervised learning (SSL) pipelines. The core idea is to encode image patches into elliptical Gaussian distributional embeddings, and use Bures-Wasserstein to calculate the (dis)similarity between encoded patches and define a Wasserstein-based attention. The authors demonstrate the performance of their proposed method across different tasks such as in-distribution generalization, out-of-distribution detection, dataset corruption, semi-supervised settings, and transfer learning to other datasets and tasks.

**Strengths:**

* The idea of stochastic token embedding and Wasserstein-based attention mechanism is interesting, and timely.

* The paper is written clearly and it is straightforward to follow.

**Weaknesses:**

* My main criticism of this paper is the experimental results.

  * First, the experimental results are limited as the experiments only focus on small-scale datasets, namely CIFAR-10 and CIFAR-100. The results will be much more conclusive if the author reports results on larger-scale datasets, e.g., imagenet, or at least datasets with larger images (e.g., mini-imagenet, or tiny-imagenet).
  * Secondly, in most of the experiments, the results either match the baselines or are only marginally superior. Additionally, the method is more than twice as costly in terms of time when compared to the baseline (~4hrs vs ~9hrs). Compounding the issue, I couldn't determine if the results are reported as an average over K runs.

  Respectfully, the results section of this paper is significantly below the standards of an average ICLR paper, and I encourage the authors to work on improving this section to increase their impact.

* There are several typos throughout the formulations that make me question the rigor of the paper.
   * Equation (4) is $W_2^2(z_1,z_2)$ and not $W_2(z_1,z_2)$. The same goes for all equations that use $W_2$.
   * Equation (10), did you mean to write: $\mathcal{L}_p-\lambda log(\sigma(-W_2^2 (z\_{out},f_z(y^+))))$?

**Questions:**

*  In Equation (10), why do you consider only positive samples? Wouldn't a generalized version using both negative and positive samples (similar to the classic works like SimCLR) be better? In other words, something like the following:

  $$-\lambda_1 log(\sigma(-W_2^2 (z_{out},f_z(y^+))))+\lambda_2 log(\sigma(-W_2^2 (z_{out},f_z(y^-)))) $$

  where if you set $\lambda_2=0$ you will recover the only positive sample case.

---

> ### Author Response · Authors · 2023-11-20
> **Response to Reviewer 67qY**
>
> We thank the reviewer for the constructive comment and for pointing out the formatting errors within the equations. We have corrected the equations correspondingly.
>
> ## Limitation of experimental results
> >First, the experimental results are limited as the experiments only focus on small-scale datasets, namely CIFAR-10 and CIFAR-100. The results will be much more conclusive if the author reports results on larger-scale datasets, e.g., imagenet, or at least datasets with larger images (e.g., mini-imagenet, or tiny-imagenet).
>
> We performed further experiments on tiny-ImageNet. These results show that our method works as expected also on larger data.
>
> ### In-distribution tiny-ImageNet experiments
> | Methods  | Acc | NLL | ECE |
> |---|---|---|---|
> | Baseline  | 65.890 |1.527 | 0.468 |
> | Ensembles k = 10  | 64.470 | 1.484 | 0.452 |
> | MC-Dropout 0.3 | 62.780 | 1.514 | 0.463 |
> | SNGP | 63.604 | 1.481 | 0.467 |
> | Sinkformer | 62.810 | 1.623 | 0.487 |
> | Our Method | 65.825 |1.466| 0.418 |
>
>
>
> ### OOD tiny-imagenet experiments
> | Methods  | AUROC |
> |---|---|
> | Baseline  | 0.501 |
> | Ensembles k = 10  | 0.511 |
> | MC-Dropout 0.3 | 0.510 |
> | SNGP | 0.543 |
> | Sinkformer | 0.505 |
> | Our Method | 0.563 |
>
> ## Significance of the results
> > Secondly, in most of the experiments, the results either match the baselines or are only marginally superior. Additionally, the method is more than twice as costly in terms of time when compared to the baseline (~4hrs vs ~9hrs). Compounding the issue, I couldn't determine if the results are reported as an average over K runs.
>
> We would like to stress that we pose the method as a novel alternative to the classic uncertainty estimation methods. While the method may be costly compared to the baseline method, our method outperformed the ensemble method, which is considered the state-of-the-art for uncertainty estimation. The computational requirements for the ensemble method are likewise 10 times costlier than our baseline method. Our method highlights a possible alternative to the commonly used state-of-the-art method that fully leverages the masking-based self-supervised learning procedure. The results are taken as an average over k = 5 runs. We will clarify this in the paper.
>
> ## Question regarding negative examples in the pre-training contrastive regularization term
> The main idea of Equation 10 was to leverage the masking procedures used in masking-based SSL frameworks such as data2vec which we use as baselines. While negative pairs may make for a more robust contrastive stochastic regularization, we opted not to sample negative pairs in the exact manner as in SimCLR to not further increase the computational overhead, which in our case is dominated by the distributional computations. The performance delivered from our method can be regarded as a lower bound performance which can potentially be optimized with a more computationally heavy negative pair sampling. Nevertheless, the inclusion and number of negative samples used in contrastive learning is also a big area of research, especially when used together with augmentations. Ash et al (2021) ([1]) pointed out that ,
> >“vision experiments are murkier with performance sometimes even being insensitive to the number of negatives”
>
> >“these experiments point to a distortion between the theoretical framework typically used to study NCE (using true positives) and, in the vision case, what is observed in practice. The core of this distortion appears to involve the use of augmentation techniques that have been highly tuned for object recognition/computer vision tasks.”
>
> [1] Ash, J. T., Goel, S., Krishnamurthy, A., & Misra, D. (2021). Investigating the Role of Negatives in Contrastive Representation Learning. (AISTATS 2022)

---

> > ### Comment · Reviewer_67qY · 2023-11-23
> > **Acknowledging Authors' Response.**
> >
> > I thank the authors for their responses and have carefully considered them. Specifically, I believe the addition of larger datasets is beneficial. However, the results still appear somewhat limited. Respectfully, the response regarding 'negative samples' did not fully address my concerns. I suggest an ablation study on the use of negative samples to strengthen the paper and make it more attractive to readers.
> >
> > I appreciate the core idea of the paper regarding stochastic token embedding, and I believe the paper has merit. However, it seems somewhat hurried, particularly in the numerical section, which could benefit from further refinement. Therefore, I maintain my original evaluation.

---

### Official Review · Reviewer_ynaf · 2023-11-02

**Soundness:** 3 good
**Presentation:** 3 good
**Contribution:** 3 good
**Rating:** 6
**Confidence:** 1

**Summary:**

This paper focuses on self-supervised learning by considering the model’s confidence and uncertainty, it proposes a new stochastic vision transformer that integrates uncertainty and distance awareness into a pipeline by a Wasserstein distance-based attention. The method is evaluated using various tasks.

**Strengths:**

The motivation is clear and convincing, the method seems nice and the methods are evaluated on various tasks.

**Weaknesses:**

n/a

**Questions:**

n/a

---

> ### Author Response · Authors · 2023-11-20
> **Response to Reviewer ynaf**
>
> We thank the reviewer ynaf for the positive recognition of the contribution of our work!

---

### Meta-Review · Area_Chair_Ztct · 2023-12-06

**Metareview:**

All the reviewers agreed that the paper is well-written, and the idea of using stochastic token embedding for uncertainty quantification is certainly interesting. The inclusion of further experiments on tiny-ImageNet during the rebuttal is also a commendable effort and was well-received. However, reviewers found that the overall contribution of the paper, while intriguing, appears somewhat limited in terms of novelty. Additionally, the practical impact in terms of performance enhancement is also somewhat constrained,.Another point raised by some reviewers is the scale of the numerical experiments conducted. While the experiments on tiny-ImageNet are a good starting point, there is a consensus that larger-scale numerical analyses would be more compelling. Such experiments would provide a more robust validation of the proposed method and could potentially increase its relevance and applicability. Based on the current state of the paper, I recommend against its acceptance for publication.

**Justification For Why Not Higher Score:**

I believe the contribution is quite small (using a previously published method for uncertainty quantification).

**Justification For Why Not Lower Score:**

N/A

---

### Decision · Program_Chairs · 2024-01-16

Reject